# Conflicts in Mitochondrial Phylogenomics of Branchiopoda, with the First Complete Mitogenome of Laevicaudata (Crustacea: Branchiopoda)

**Xiaoyan Sun and Jinhui Cheng ***

State Key Laboratory of Palaeobiology and Stratigraphy, Nanjing Institute of Geology and Palaeontology and Center for Excellence in Life and Palaeoenvironment, Chinese Academy of Sciences, 39 Beijing Eastroad, Nanjing 210008, China
* Correspondence: jhcheng@nigpas.ac.cn

**Abstract:** Conflicting phylogenetic signals are pervasive across genomes. The potential impact of such systematic biases may be reduced by phylogenetic approaches accommodating for heterogeneity or by the exclusive use of homoplastic sites in the datasets. Here, we present the complete mitogenome of *Lynceus grossipedia* as the first representative of the suborder Laevicaudata. We employed a phylogenomic approach on the mitogenomic datasets representing all major branchiopod groups to identify the presence of conflicts and concordance across the phylogeny. We found pervasive phylogenetic conflicts at the base of Diplostraca. The homogeneity of the substitution pattern tests and posterior predictive tests revealed a high degree of compositional heterogeneity among branchiopod mitogenomes at both the nucleotide and amino acid levels, which biased the phylogenetic inference. Our results suggest that Laevicaudata as the basal clade of Phyllopoda was most likely an artifact caused by compositional heterogeneity and conflicting phylogenetic signal. We demonstrated that the exclusive use of homoplastic site methods combining the application of site-heterogeneous models produced correct phylogenetic estimates of the higher-level relationships among branchiopods.

**Keywords:** Laevicaudata; mitochondrial genome; compositional heterogeneity; phylogenetic signal; model violation

## 1. Introduction

With an increasing number of mitogenomes being sequenced and various methodological advances, mitogenomic data have been successfully utilized to improve phylogenetic reconstructions across a wide range of taxa [1–3]. The large amount of available mitogenomic data has reduced the stochastic error (sampling error) on phylogenetic inference. Nevertheless, deep relationships between Arthropoda at the interordinal or intraordinal level have not been fully resolved, resulting in topologies with high support frequently conflicting with morphological and nuclear phylogenies [4–6]. Such strong support but incorrect phylogenies represent systematic errors which can be traced back to homoplastic characteristics in datasets and model violations [7,8]. Substitutional saturation was the most frequently discussed cause of homoplasy in nucleotide gene data [9]. Most substitution models assume compositional homogeneity (stationary), but nucleotide and protein sequences might also exhibit nonstationarity, which strongly violates the assumptions of the stationary models [10,11]. The most common source of model violations are compositional heterogeneity and rate heterogeneity among lineages [12–14].

Suborder Laevicaudata Linder, 1945 (Branchiopoda: Diplostraca), or smooth clam shrimp, is a unique group of essentially benthic micro-crustaceans. Among branchiopods, Laevicaudata can be recognized by a usual body length less than 7 mm, a bivalved carapce, a proportionally large head, bearing a row of large teeth on the mandibular molar surface and having laminae abdominalis supporting egg clusters [15–19]. Laevicaudata currently

comprises about 42 valid species worldwide except in Antarctica [15,16,18,20–24]. Laevicaudata, including only one family, Lynceidae Baird, 1845, is composed of three genera: *Lynceus* Müller, 1776; *Lynceiopsis* Daday, 1912; and *Paralimnetis* Gurney, 1931. They are distinguished by the shape and size of male claspers, characteristic modified second thoracopods, and rostrum [15–17,20], and *Lynceus* represents nearly 90% of all laevicaudatan species diversity [17,18,25].

The higher-level relationships of Branchiopoda are well resolved based on both morphological and molecular data [26–32]. The class Branchiopoda is divided into two superorders, Anostraca and Phyllopoda, and three extant orders, Anostraca, Notostraca and Diplostraca. Laevicaudata and Onychocaudata (Spinicaudata, Cyclestherida and Cladocera) are sister clades, forming the order Diplostraca, which, together with the order Notostraca, belongs to the superorder Phyllopoda [26–28]. It is widely accepted that Laevicaudata and Onychocaudata form a well-defined monophyletic order called Diplostraca [26–30]. Mitogenomes have substantially aided in estimating phylogenetic relationships within clades of Branchiopoda such as Anostraca, Notostraca and Cladocera [33–36], but limited success has been achieved in resolving the deep relationship between Laevicaudata and Onychocaudata. The large amount of available mitogenomic data provides a high phylogenetic resolution of the relationships among Cladocera. However, phylogenetic estimates thus far have resulted in strong support but incorrect phylogenies for Diplostraca, even using a site-heterogeneous mixture model [37], which might indicate a systematic bias arising from model violation.

In order to investigate the phylogenetic signal contained in the mitogenomes for major groups of Branchiopoda, we sequenced and annotated the complete mitogenome of *L. grossipedia* (Lynceidae) as the first complete mitogenome of the suborder Laevicaudata. The aim of this study is to account for the heterogeneity of sequences and dissect phylogenetic signals in the mitogenomic dataset, coupling them with available mitogenomes of Branchiopoda from GenBank (http://www.ncbi.nlm.nih.gov, accessed on 28 September 2022). We evaluated three methods for diminishing the non-phylogenetic signal concerning their effectiveness in reducing model violations and their influence on the phylogenetic reconstruction. We also compared the results of phylogenetic reconstruction with different approaches based on different datasets. Finally, we conducted four-cluster likelihood mapping analyses (FcLM) to evaluate and visualize the phylogenetic signal in each dataset.

Herein, using the higher-level relationships described above as correct topology, we presented the pervasiveness of phylogenetic conflicts at the base of Diplostraca. The results uncovered hitherto unrecognized nonphylogenetic signals as the artifactual origin of the conflicting topologies. The verification methods explicitly taking systematic bias into consideration consistently supported the monophyly of the Diplostraca.

## 2. Materials and Methods

### 2.1. Sample Collection and DNA Extraction

*L. grossipedia* was collected from Chengde of Hebei Province, China (116°10′ E, 41°28′ N). All samples were morphologically identified and preserved in 95% ethanol at −20 °C for DNA extraction. All specimens and vouchers (No. LGPHECD01-11) were deposited in the State Key Laboratory of Palaeobiology and Stratigraphy, Nanjing Institute of Geology and Palaeontology, Chinese Academy of Sciences, Nanjing, China. Total DNA was extracted using the DNeasy tissue kit (Qiagen, Hilden, Germany) following the manufacturer's instructions.

### 2.2. PCR Amplification, Sequencing, Sequence Assembly, and Gene Annotation

The mitochondrial genome was amplified by Polymerase Chain Reaction (PCR) using 14 primer pairs (Supplementary Table S1). Amplification reactions and sequencing were performed according to the previously described method [38], and assembling of mtDNA fragments, annotation of mitogenome, and comparison followed the procedure of Sun and Cheng [39]. Overlapped mtDNA fragments were assembled into contigs using

BioEdit 7.0.9.0 [40]. Sequence annotation and accurate boundary determination of PCGs and rRNA genes were first performed by the NCBI's web interface for BLAST [41] and then by alignment with the homologous genes from other released sequences of Branchiopoda. Miotochondrial tRNA genes and their secondary structures were identified by a combination of MITOS online software [42] and tRNAscan-SE 1.2.1 online software [43].

### 2.3. Sequence Alignment and Substitutional Saturation Test

Mitochondrial genomes of 44 relevant taxa were retrieved from GenBank (http://www.ncbi.nlm.nih.gov, accessed on 28 September 2022), together with our newly generated Laevicaudata mitogenome, resulting in a dataset (Supplementary Table S2). The dataset is composed of 42 ingroup species representing 16 families and 3 orders of Branchiopoda: Diplostraca (5 species), Notostraca (3 species), and Anostraca (22 species). *Hutchinsoniella macracantha* Sanders, 1955 (Cephalocarida: Hutchinsoniellidae) and *Squilla biformis* Bigelow, 1891 (Malacostraca: Squillidae) were selected as outgroups. The amino acid sequences of 13 protein-coding genes (PCGs) were aligned using MUSCLE implemented in MEGA X [44]. The corresponding nucleotide sequences of each PCG were aligned by the aligned amino acid sequences implemented in DAMBE 6 [45].

We estimated saturation for each PCG, and for the three codon positions using DAMBE 6 [45], which determined an "index of substitution saturation" ($I_{ss}$) based on the notion of entropy in information theory. We excluded partitions or genes which showed significant nucleotide saturation from phylogenetic analyses. The non-synonymous substitution rate ($K_a$) for each taxon was calculated in comparison with the outgroup using DAMBE 6 [45].

### 2.4. Analyses of Sequence Heterogeneity and Phylogenetic Signal Dissection

We calculated the base composition of each taxon for each PCG and compared the AT% for each gene among the branchiopod species included in this study. The compositional diversity of amino acids of 13 PCGs across branchiopod suborders was obtained by calculating the frequency of four amino acids which were encoded by GC-rich codons (glycine, alanine, arginine and proline; GARP). The homogeneity of substitution pattern ($I_D$ test) for each gene was estimated using a Monte-Carlo method with 1000 replicates implemented in MEGA X [44]. The null hypothesis that sequences have evolved with the same pattern of substitution was rejected at $\alpha < 0.01$. We also evaluated the compositional heterogeneity in each of the 13 mitochondrial proteins separately by performing posterior predictive analysis (PPA) with the global test statistic as implemented in PhyloBayes 4.1c [46].

A significant conflict between the branchiopod phylogenies is the placement of Laevicaudata. To resolve the deep relationship between Laevicaudata and Onychocaudata and to address the sources of deep phylogenetic conflict, we divided taxa into four clades: (1) Anostraca; (2) Notostraca; (3) Laevicaudata; and (4) Onychocaudata.

Three methods for reducing the nonphylogenetic signals were conducted: (1) exclusion of the genes with the most strongly deviating composition according to the AT% of *L. grossipedia*; (2) exclusion of the proteins with a significant model violation according to posterior predictive analysis (PPA); and (3) removal of fast-evolving sites. To evaluate the key phylogenetic splits and visualize the phylogenetic content of datasets, we conducted four-cluster likelihood mapping analyses (FcLM) using both nucleotide and amino acid datasets as implemented in TreePuzzle v5.3 [47]. We preferred the topology of the currently accepted relationships within Branchiopoda: (((Laevicaudata, Onychocaudata), Notostraca), Anostraca).

### 2.5. Phylogenetic Analysis

Five datasets were used for phylogenetic analyses: (1) 13 protein-coding genes without the third codon positions (the PCG12 matrix; 7587 bp); (2) amino acid sequences of 13 PCGs (the PAA matrix; 3796 aa); (3) a concatenated nucleotide sequence alignment of the first and the second codon positions of six PCGs including *cox1*, *cox2*, *cox3*, *cytb*, *atp6* and *nad3* (the Pnuc6 matix; 3464 bp); (4) a concatenated amino acid sequence alignment of seven PCGs

except *atp8*, *nad2*, *nad4*, *nad4l*, *nad5* and *nad6* (the Paa7 matrix; 2043 aa); (5) a concatenated amino acid sequence alignment removing fast-evolving sites from Paa7 matrix (the Paas7 matrix; 1094 aa).

### 2.5.1. Phylogenetic Analyses under Site-Homogeneous Models

In order to compare the results of phylogenetic inference from different evolutionary models, phylogenetic analyses of four datasets (PCG12, PAA, Pnuc6 and Paa7) were first carried out under site-homogeneous models implemented in RAxML 2.2.3 [48] for maximum likelihood inference (ML) and Bayesian inference using MrBayes 3.2 [49]. Because highly heterogeneous sequence divergence was present across branchiopod clades, and applying standard homogenous models might prompt inaccurate inferences, we did not apply this method to the Paas7 matrix. The best-fit model for the nucleotide dataset (Pnuc6) according to the Akaike Information Criterion (AIC) was determined using jModelTest version 0.1.1 [50], and ProtTest 3 [51] was used for the amino acid dataset (Paa7). The best selected partition schemes and models of two datasets were listed in Supplementary Table S3. For the ML analyses, branch support of two datasets was estimated using the rapid bootstrap method in RAxML with 1000 replicates. Analyses with the software MrBayes were conducted in two simultaneous runs, each with four chains, for 10 million generations, and trees being sampled every 1000 generations. The first 25% were discarded as burn-in, and the remaining trees were used to calculate Bayesian posterior probability (BPP) values. Values of the Potential Scale Reduction Factor (PSRF) approaching 1.0 suggested that the runs reached convergence.

### 2.5.2. Phylogenetic Analyses under Site-Heterogeneous Models

Substitution saturation is recognized as one of the primary obstacles for deep phylogenetic inference, and removing sites that have experienced multiple substitutions would make for erratic phylogenetic estimates [52]. In order to correctly retrieve the phylogenetic signals of pattern-heterogeneity from mitogenomic sequence data, we performed Bayesian inference analyses under the site-heterogeneous model CAT + GTR for three datasets (Pnuc6, Paa7, and Paas7), as implemented in PhyloBayes 4.1c [46]. Two independent chains of 5000 cycles were run for each analysis, with one point every five samples. The initial 1000 trees sampled in each MCMC run were discarded as burn-in after checking for convergence using bpcomp (max_diff < 0.3). The 50% majority-rule consensus tree and the associated posterior probabilities (PPs) were then computed using all chains.

Bayesian cross-validation [53] was used to compare the fit of site-homogeneous (LG and GTR) and site-heterogeneous (CAT-mtREV and CAT-GTR) models as implemented in PhyloBayes 4.1c [46]. Ten replicates were conducted, 1100 sampling cycles were run and the first 100 samples were discarded as burn-in. Fast-evolving sites for Paa7 matrix were identified using the discrete gamma rate category to which they belong using TreePuzzle v5.3 [47], and the sites belonging to the most rapidly evolving gamma category were removed.

## 3. Results

### 3.1. Characteristics of L. grossipedia Mitogenome

The complete mitogenome of *L. grossipedia* is 15,023 bp in length (GenBank accession number: OP746066). This is the first completely sequenced mitogenome in the order Laevicaudata. All of the 37 typical animal mitochondrial genes were identified, consisting of 13 PCGs, 22 tRNAs, two mitochondrial ribosomal RNAs (rrnS and rrnL) and a putative control region (Table 1). Twenty-three genes were encoded by the majority strand (J-strand) and fourteen by the minority strand (N-strand). Gene arrangement of the branchiopod mitogenomes was considered to be rather well-conserved, although several events of translocation, inversion, tandem duplication and random loss have occurred [33,34]. We found two gene rearrangement phenomena in the mitogenome of *L. grossipedia*: (1) the local inversion of *trnI*, and (2) the remote inversion of *trnL₁*. The latter observed at the *nad1–rrnL*

junction was the dominant gene rearrangement event in Branchiopoda. In addition to the control region, the mitogenome of *L. grossipedia* had 181 bp of intergenic nucleotides in 13 different locations, with intergenic spacer lengths ranging from 1 to 63 bp. The longest intergenic spacer was located between *trnG* and *nad3* (Table 1). In the *L. grossipedia* mitogenome, ATN codons initiated all PCGs. Six PCGs used TAA/TAG as the termination codons, while truncated termination codons (T) was observed in the other seven genes (Table 1).

**Table 1.** Annotation of the mitochondrial genome of *L. grossipedia*.

| Gene | GenBank Position no. | Size (nts) | Strand [a] | Start Codon | Stop Codon | Anticodon | IGN [b] |
|---|---|---|---|---|---|---|---|
| *trnI* | 1–64 | 64 | - | | | GAT | 46 |
| *trnQ* | 111–179 | 69 | - | | | TTG | 9 |
| *trnM* | 189–253 | 65 | + | | | CAT | 12 |
| *nd2* | 266–1216 | 951 | + | ATA | TAG | | −2 |
| *trnW* | 1215–1277 | 63 | + | | | TCA | −1 |
| *trnC* | 1277–1339 | 63 | - | | | GCA | 0 |
| *trnY* | 1340–1405 | 66 | - | | | GTA | −5 |
| *cox1* | 1401–2939 | 1539 | + | ATA | TAA | | 2 |
| *trnL1*-CUN | 2942–3006 | 65 | + | | | TAG | 1 |
| *trnL2*-UUR | 3008–3069 | 62 | + | | | TAA | 19 |
| *cox2* | 3089–3770 | 682 | + | ATT | T | | 1 |
| *trnK* | 3771–3835 | 65 | + | | | CTT | 0 |
| *trnD* | 3836–3898 | 63 | + | | | GTC | 0 |
| *atp8* | 3899–4060 | 162 | + | ATT | TAA | | −4 |
| *atp6* | 4057–4720 | 664 | + | ATA | T | | 0 |
| *cox3* | 4721–5508 | 788 | + | ATG | T | | −1 |
| *trnG* | 5509–5569 | 61 | + | | | TCC | 63 |
| *nd3* | 5633–5990 | 358 | + | ATA | T | | 3 |
| *trnA* | 5991–6052 | 62 | + | | | TGC | 14 |
| *trnR* | 6067–6126 | 60 | + | | | TCG | −3 |
| *trnN* | 6124–6187 | 64 | + | | | GTT | 0 |
| *trnS1*-AGN | 6188–6244 | 57 | + | | | GCT | 0 |
| *trnE* | 6245–6307 | 63 | + | | | TTC | 0 |
| *trnF* | 6308–6369 | 62 | - | | | GAA | 0 |
| *nd5* | 6370–8041 | 1672 | - | ATT | T | | 0 |
| *trnH* | 8042–8103 | 62 | - | | | GTG | 0 |
| *nd4* | 8104–9403 | 1300 | - | ATG | T | | −1 |
| *nd4L* | 9403–9690 | 295 | - | ATG | TAA | | 5 |
| *trnT* | 9696–9757 | 62 | + | | | TGT | 0 |
| *trnP* | 9758–9820 | 63 | - | | | TGG | 2 |
| *nd6* | 9823–10,303 | 481 | + | ATT | T | | 0 |
| *cytb* | 10,304–11,434 | 1131 | + | ATG | TAA | | −2 |
| *trnS2*-UCN | 11,433–11,499 | 67 | + | | | TGA | 4 |
| *nd1* | 11,504–12,415 | 912 | - | ATT | TAA | | 0 |
| *rrnL* | 12,416–13,738 | 1323 | - | | | | 0 |
| *trnV* | 13,739–13,806 | 68 | - | | | TAC | 0 |
| *rrnS* | 13,807–14,586 | 780 | - | | | | 0 |
| Control region | 14,589–15,023 | 437 | + | | | | 0 |

[a] Plus strand (+)/mius strand (-); [b] Number of intergenic nucleotides. Numbers of IGN indicate non-coding nucleotides between genes (positive values) or gene overlap (negative values).

For *L. grossipedia*, the AT content of the complete genome, PCGs, rRNA, tRNA and control region were greater than those of *Leptestheria brevirostris* Barnard, 1924, 75%, 73.6%, 76.8%, 75.8% and 81%, respectively (Table 2). The highest AT content occurs in the third codon position of PCGs (84.3%). *Atp8* had a very high AT content (83.7%), while the lowest AT content was found in *cox1* (66.8%). The AT contents of *L. grossipedia* were the highest when compared with other species of Branchiopoda, showing an obvious AT mutation

bias. In most branchiopods, the mitogenome has a positive AT-skew and negative GC-skew [37,39]. *L. grossipedia* and *Leptestheria brevirostris* exhibited a similar pattern (Table 2). The nucleotide skew statistics for the mitochondrial genomes of *L. grossipedia* analysed in the present study also indicated the following: (1) the AT-skew of all PCGs was negative ($-0.32 \sim -0.07$); (2) the GC-skew was positive and the AT-skew of each PCG on the minority strand was negative, whereas both the GC-skew and AT-skew of each PCG on the majority strand were negative (except *cox1*) (Table 2).

**Table 2.** Nucleotide composition and skewness levels of *L. grossipedia* (Laevicaudata)/*Leptestheria brevirostris* (Spinicaudata).

| Regions | Nucleotide Composition (%) | | | | | AT-Skew | GC-Skew |
|---|---|---|---|---|---|---|---|
| | T (U) | C | A | G | A + T | | |
| Whole genome | 36.8/31.9 | 15.3/22.4 | 38.1/26.5 | 9.9/19.2 | 75.0/59.5 | 0.02/−0.09 | −0.21/−0.08 |
| PCGs | 43.8/35.4 | 12.5/22.6 | 29.8/22.0 | 13.8/20.0 | 73.6/57.4 | −0.19/−0.23 | 0.05/−0.06 |
| 1st codon position | 35.4/29.0 | 12.6/20.5 | 32.0/25.6 | 20.0/25.1 | 67.4/54.4 | −0.05/−0.06 | 0.24/0.10 |
| 2nd codon position | 48.6/44.0 | 16.9/22.2 | 20.5/17.3 | 14.0/16.5 | 69.2/61.4 | −0.41/−0.43 | −0.10/−0.15 |
| 3rd codon position | 47.4/33.0 | 8.1/25.1 | 37.0/23.0 | 7.6/18.5 | 84.3/56.0 | −0.12/−0.19 | −0.03/−0.15 |
| rRNA | 39.3/27.3 | 7.6/16.3 | 37.5/34.8 | 15.7/21.53 | 76.8/62.1 | −0.02/0.12 | 0.35/0.14 |
| tRNA | 36.7/31.3 | 10.7/16.6 | 39.1/30.1 | 13.5/22.1 | 75.8/61.3 | 0.03/−0.02 | 0.12/0.14 |
| *atp6* | 42.0/35.1 | 16.9/25.2 | 31.5/20.1 | 9.6/19.5 | 73.5/55.3 | −0.14/−0.27 | −0.28/−0.13 |
| *atp8* | 45.6/33.3 | 12.2/28.8 | 38.1/23.1 | 4.1/14.7 | 83.7/56.4 | −0.09/−0.18 | −0.5/−0.32 |
| *cox1* | 39.2/33.5 | 16.7/23.7 | 27.6/20.5 | 16.5/22.2 | 66.8/54.1 | −0.17/−0.24 | −0.01/−0.03 |
| *cox2* | 38.2/30.8 | 16.6/24.7 | 32.9/22.9 | 12.3/21.6 | 71.1/53.7 | −0.07/−0.15 | −0.15/−0.07 |
| *cox3* | 39.5/37.1 | 17.1/22.3 | 29.8/19.6 | 13.7/21.0 | 69.2/56.7 | −0.14/−0.31 | 0.11/−0.03 |
| *cytb* | 39.5/34.3 | 16.6/24.4 | 31.6/20.6 | 12.4/20.7 | 71.0/54.9 | −0.11/−0.25 | −0.14/−0.08 |
| *nad1* | 48.7/36.4 | 8.5/18.6 | 26.2/24.1 | 16.6/20.8 | 74.9/60.6 | −0.30/−0.20 | 0.32/0.06 |
| *nad2* | 46.3/40.0 | 11.4/27.1 | 31.8/17.6 | 10.4/15.4 | 78.2/57.6 | −0.19/−0.39 | −0.05/−0.28 |
| *nad3* | 43.2/40.2 | 15.3/19.9 | 30.2/18.8 | 11.3/21.1 | 73.4/59.0 | −0.18/−0.36 | −0.15/0.03 |
| *nad4* | 49.8/36.5 | 7.4/21.1 | 27.9/22.5 | 14.9/20.8 | 77.8/59.0 | −0.28/−0.24 | 0.34/0.02 |
| *nad4L* | 52.2/34.3 | 3.4/19.9 | 26.8/24.9 | 17.5/20.9 | 79.0/59.3 | −0.32/−0.16 | 0.67/0.02 |
| *nad5* | 44.5/32.3 | 8.1/20.6 | 31.8/26.6 | 15.6/20.5 | 76.2/58.8 | −0.17/0.10 | 0.31/0.00 |
| *nad6* | 45.8/39.1 | 13.1/25.7 | 34.2/20.1 | 6.9/15.1 | 80.0/59.2 | −0.15/−0.32 | −0.31/−0.26 |

### 3.2. Levels of Substitutional Saturation and Heterogeneous Sequence Divergence within Branchiopod Mitogenomes

The third codon positions were saturated for all genes, and about half of the first and second codon position also showed significant levels of saturation (Table S4). Therefore, they were not considered for further phylogenetic analyses.

The value of $K_a$ was low for Notostraca (0.24~0.25), Spinicaudata (0.26~0.27) and Cladocera (0.25~0.27), but generally higher for Anostraca (0.32~0.38) and Laevicaudata (0.32 ± 0.01), which suggested that Anostraca and Laevicaudata had relatively higher evolutionary rates among Branchiopoda. We analysed the compositional heterogeneity of both the nucleotides and amino acids of 13 PCGs across branchiopod suborders. There was considerable variation in the AT content of mitogenomes within branchiopods (47.8%~83.7%), and Laevicaudata had the highest AT% by a high margin (Table 3). There was considerable variation in GC-encoding GARP amino acids of the mitochondrial genome within Branchiopoda (range: 14.37%~18.78%; mean: 17.66%; standard deviation: 1.24), and Laevicaudata had the lowest GARP%. Our observation showed a high degree of compositional heterogeneity among branchiopod mitogenomes in both nucleotide and amino acid level, which led to systematic error in phylogenetic analyses [11,54–56]. To test the homogeneity of substitution pattern, we made 903 pairwise comparisons to calculate the $I_D$. When we compared the concatenated 13 PCGs, 719 comparisons had a statistically significant heterogeneous substitution pattern, suggesting that the substitution pattern evolved multiple times. The $I_D$ test on each 13 PCGs also suggested a high level of variation in the substitution patterns among different genes (Table 3). The null hypothesis that sequences

have evolved with the same pattern of substitution was rejected ($\alpha < 0.01$), although a correlation was observed between the level of variation in the substitution patterns and the gene lengths.

**Table 3.** $I_D$ test summary and AT% on individual mitochondrial genes.

| Gene | Number of Comparisons with Significant Heterogeneity | Proportion of Significant Heterogeneity (%) | AT% [a] | | | |
|------|------|------|------|------|------|------|
| | | | Max | Mean | Min | LG [b] |
| *nd2* | 554 | 61.4 | 78.2 | 68.6 | 57.2 | 78.2 |
| *cox1* | 588 | 65.1 | 66.8 | 61.7 | 54.1 | 66.8 |
| *cox2* | 368 | 40.8 | 71.1 | 64.2 | 53.7 | 71.1 |
| *atp8* | 123 | 13.6 | 83.7 | 70.2 | 47.8 | 83.7 |
| *atp6* | 435 | 48.2 | 73.5 | 64.8 | 54.6 | 73.5 |
| *cox3* | 597 | 66.1 | 69.2 | 62.3 | 51.2 | 69.2 |
| *nd3* | 350 | 38.8 | 76.8 | 68.7 | 56.8 | 73.5 |
| *nd5* | 710 | 78.6 | 76.2 | 66.5 | 54.8 | 76.2 |
| *nd4* | 692 | 76.6 | 77.8 | 67.0 | 55.1 | 77.8 |
| *nd4l* | 318 | 35.2 | 79.0 | 68.8 | 59.3 | 79.0 |
| *nd6* | 334 | 37.0 | 80.0 | 69.9 | 55.7 | 80.0 |
| *cytb* | 569 | 63.0 | 71.0 | 62.9 | 54.9 | 71.0 |
| *nd1* | 598 | 66.2 | 74.9 | 65.6 | 56.9 | 74.9 |

Note: For each gene, a total of 903 pairwise comparisons are made and shown. The null hypothesis that sequences have evolved with the same pattern of substitution is rejected ($\alpha < 0.01$). AT% [a]: AT% of each 13 PCGs of Branchiopoda; LG [b]: AT% of each 13 PCGs of *L. grossipedia*.

### 3.3. Phylogenetic Analyses Using Standard Homogeneous Models

Homogeneous analyses of either nt or aa data yielded maximal support for the monophyly of Anostraca and Phyllopoda ($ML_{nt\&aa}$-BS = 100%; Figure 1). Phyllopoda included four major groups (Cladocera, Spinicaudata, Laevicaudata and Notostraca), and Laevicaudata was resolved as a sister to the remaining phyllopods. However, the relationships among these four groups differed based on different datasets: monophyletic Notostraca was resolved as a sister to Onychocaudata when inferences were drawn from nucleotide data ($ML_{nt\&aa}$-BS = 95%; Figure 2a), whereas Notostraca occupied a sister position to Spinicaudata based on the amino acid data ($ML_{nt\&aa}$-BS = 62%; Figure 2b), which were consistent with the four-cluster likelihood mapping analyses (nt: 57.1% and aa: 72.1%, Figure 2c,d). These results, based on site-homogeneous analyses, were congruent with previous mitogenomes analyses [37], but not consistent with the currently accepted sister group relationship between Laevicaudata and Onychocaudata [26,29,33]. The conflict was not resolved by the Bayesian and ML analyses under site-homogenous models with a partitioning scheme for both Pnuc6 and Paa7 datasets, each matrix supporting similar trees (Figure S1) to those presented in Figure 1.

### 3.4. Reducing Compositional Heterogeneity in Sequence Data

Phylogenetic analyses of the individual mitochondrial genes and proteins and PPA test demonstrated that both the nucleotide composition of all 13 PCGs and the amino acid composition of six among the 13 mitochondrial proteins violated the assumptions of the CAT model (Table 4), indicating that compositional bias was usually a genome-wide phenomenon [54].

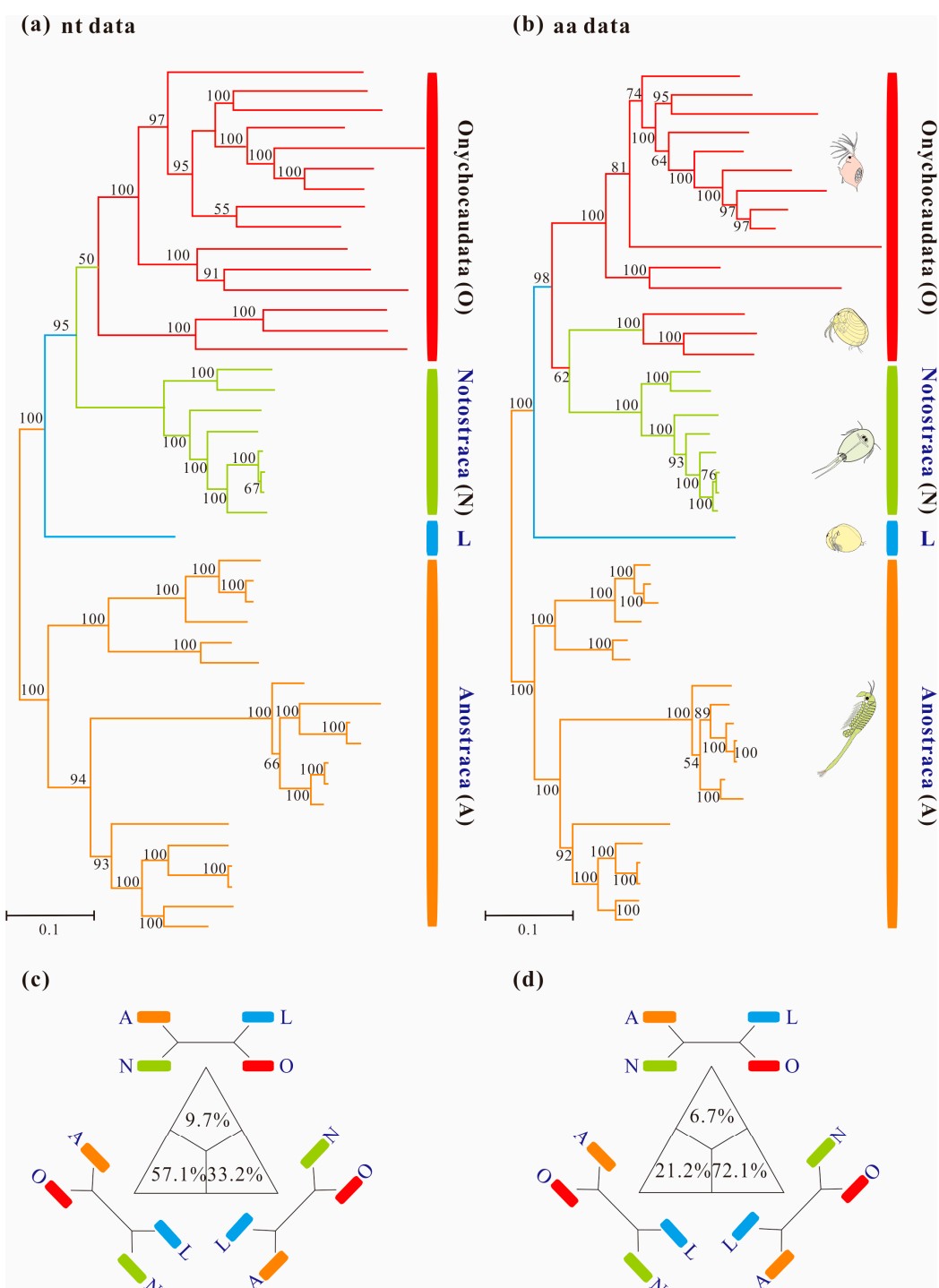

**Figure 1.** Maximum-likelihood phylograms of Branchiopoda based on concatenation of 13 mito-chondrial genes under site-homogeneous models obtained with RAxML: (**a**) PCG12 matrix and (**b**) PAA matrix. Major groups are labeled and each group is indicated with a representative line drawing. Nodal supports are bootstrap values. The trees of Branchiopoda between the nucleotide and amino acid datasets exhibit incongruence by the monophyly of Onychocaudata. (**c**) and (**d**) are results based on the PCG12 and PAA datasets respectively using the four-cluster likelihood mapping analyses. For each matrix, the sequences are divided into four clades: (**a**) Anostraca; (**b**) Notostraca; (**c**) Laevicaudata; (**d**) Onychocaudata. The corners of the triangles represent the three alternative topologies.

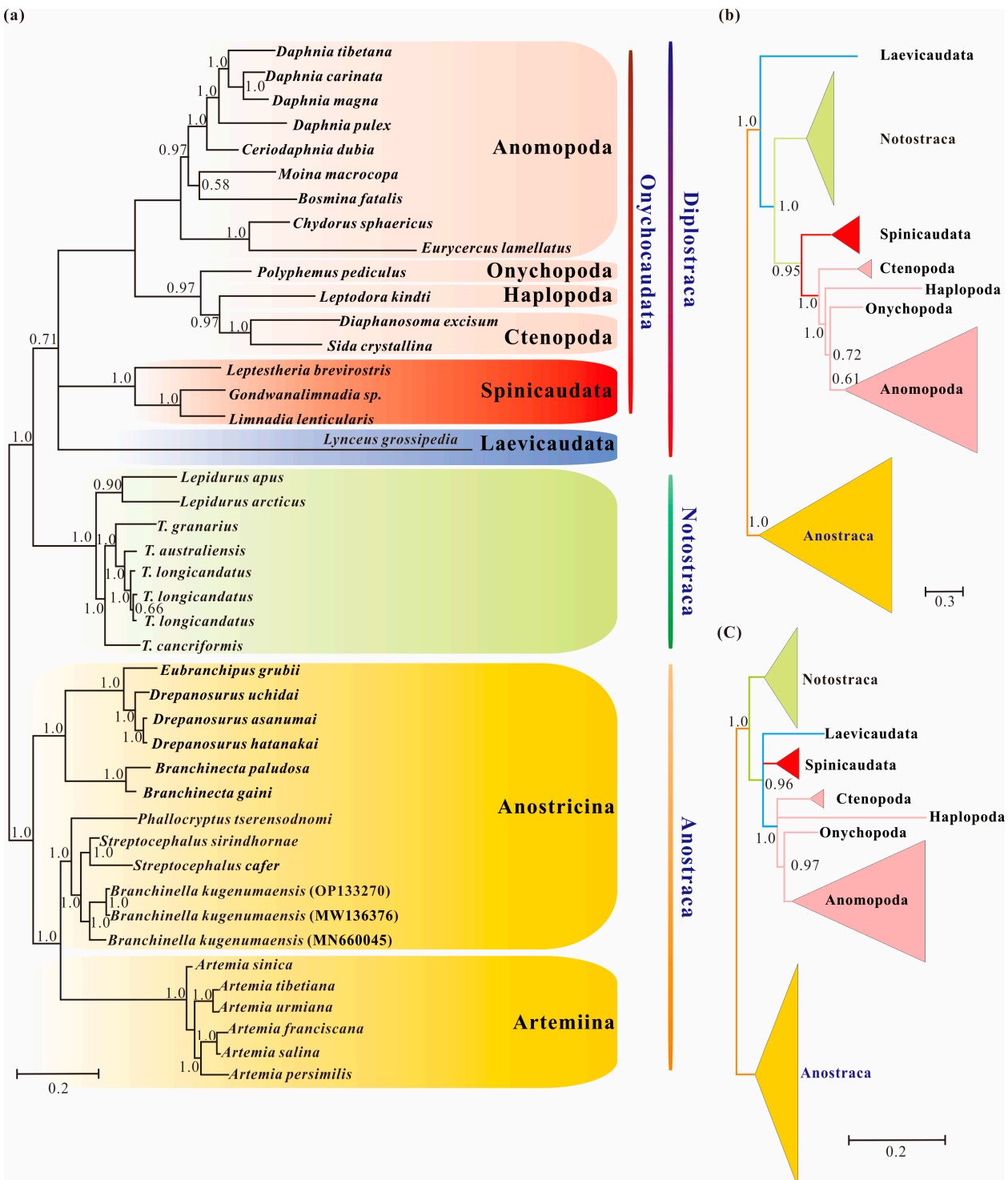

**Figure 2.** Branchiopod phylogenies inferred from the datasets of Pnuc6, Paa7 and Paas7 under the site-heterogeneity models obtained with PhyloBayes: (**a**) Bayesian tree from the dataset of Pnuc6; (**b**) Bayesian tree from the dataset of Paa7; (**c**) Bayesian tree from the dataset of Paas7. The Bayesian trees based on the datasets of Paa7 and Paas7 are shown in b and c as a schematic version with some lineages collapsed for clarity. Supports at nodes are Bayesian posterior probabilities.

**Table 4.** Results of Posterior Predictive Test on individual mitochondrial genes.

| Gene/Datasets | Nucleotide Data Sets | | | Amino Acid Data Sets | | |
|---|---|---|---|---|---|---|
| | Z Score | p Score | NDT * | Z Score | p Score | NDT * |
| *nd2* | 6.42 | 0.00 | 31 | 3.50 | 0.00 | 18 |
| *cox1* | 5.20 | 0.02 | 32 | 0.37 | 0.24 | 6 |
| *cox2* | 4.82 | 0.03 | 18 | 0.01 | 0.42 | 0 |
| *atp8* | 5.67 | 0.00 | 11 | 2.11 | 0.03 | 14 |
| *atp6* | 4.67 | 0.00 | 25 | 0.26 | 0.30 | 4 |
| *cox3* | 4.31 | 0.02 | 31 | −0.89 | 0.81 | 2 |
| *nd3* | 6.05 | 0.01 | 20 | 0.41 | 0.29 | 3 |
| *nd5* | 13.10 | 0.00 | 33 | 5.79 | 0.00 | 16 |
| *nd4* | 6.81 | 0.00 | 39 | 4.34 | 0.00 | 10 |
| *nd4l* | 7.12 | 0.00 | 26 | 2.04 | 0.05 | 5 |
| *nd6* | 5.67 | 0.01 | 24 | 2.15 | 0.05 | 5 |
| *cytb* | 6.59 | 0.00 | 26 | 0.92 | 0.19 | 3 |
| *nd1* | 5.13 | 0.00 | 35 | 1.28 | 0.09 | 7 |
| 13 PCGs | 5.60 | 0.00 | 39 | 13.18 | 0.00 | 33 |

NDT *: Number of taxa with significantly deviating composition.

The sequence heterogeneity analysis showed that Laevicaudata exhibited significantly higher heterogeneity than the other branchiopods. Laevicaudata being resolved as the basal clade of Phyllopoda with high support was most likely due to artifactual phylogenetic inferences, probably resulting from the high degree of heterogeneity. If we excluded Laevicaudata, no inferences can be made about the relationships of this taxon. Accordingly, we applied three exclusive uses of homoplastic sites methods to reduce the potential impact of compositional heterogeneity on phylogenetic inference. Using the concatenated sequences of the 13 PCGs or 13 mitochondrial proteins for phylogenetic inference was proven to be not impactful in reconciling model misspecification (Table 4). When only seven mitochondrial proteins or six PCGs with the lowest Z scores were used for the phylogenetic analyses, the Z scores were reduced, but the compositional heterogeneity was still significant (Z = 7.27 for 7 proteins; Z = 5.00 for 6 PCGs). However, when only the Paas7 matrix was used for phylogenetic analyses, the CAT model was no longer violated (*p* = 0.09, Z = −1.48). Therefore, the compositional heterogeneity in the concatenated sequences of mitochondrial proteins could be reduced to a degree that the CAT model was no longer violated.

*3.5. Phylogenetic Results under Heterogeneous Model*

Nonstationary heterogeneous composition models, which account for compositional heterogeneity among lineages, have been manifested to control systematic errors in tree reconstruction [10,57]. The results of Bayesian cross-validation tests showed that: (1) The CAT-GTR+$\Gamma_4$ mixture model offered a better fit to the data compared with GTR+$\Gamma_4$ (2$\Delta$lnL = 430 ± 48), and (2) the CAT-mtREV+$\Gamma_4$ mixture model was better, compared with LG + $\Gamma_4$ (2$\Delta$lnL = 16086 ± 1546).

Bayesian inference from the Pnuc6 dataset under a site-heterogeneous model recovered the monophyly of Diplostraca, but with low probability (BI$_{nt}$-PP = 0.71; Figure 2a). The result indicated that the high support for Laevicaudata as the earliest branch of Phyllopoda under site-homogeneous models was partly due to among-lineage compositional bias. In contrast, the analysis of the Paa7 matrix using the CAT-mtREV+$\Gamma_4$ mixture model model did not support the monophyly of Diplostraca, and the result supported Laevicaudata as a sister to the rest of Phyllopoda (Figure 2b), as did site-composition homogeneous

models. When we removed fast-evolving sites (46.45%) from Paa7 matrix, the monophyly of Diplostraca was recovered with high support (BI$_{nt}$-PP = 0.96; Figure 2c), confirming that the removal of the fast-evolving positions increased the ratio of phylogenetic to non-phylogenetic signal [58]. This observation also implied that compositional heterogeneity and fast-evolving positions in the amino acid datasets were two sources of phylogenetic artifacts and nonstationary heterogeneous composition model showed significant improvements over site-homogenous models in the phylogenetic reconstruction.

## 4. Discussion

### 4.1. Pervasiveness of Phylogenetic Conflicts

In this study, several datasets were utilized to test basal relationships of Diplostraca and to compare the results of phylogenetic inference from different evolutionary models. Nevertheless, the standard phylogenetic methods consistently failed to uncover the correct phylogeny (Figure 1 and Figure S1). High nodal supports concealed the pervasiveness of phylogenetic conflicts. The non-monophyly of Diplostraca supported by these analyses was an artifactual result overturning a key relationship supported by morphological cladistic studies [26–28,31] and phylogenomic analyses [29,30,32,59]. The monophyly of Diplostraca was supported based on the list of supporting synapomorphies, such as bivalved carapaces in adults, larvae with small and budlike first antennae, highly modified first male thoraco-pod pair for clasping females and trunk limb exopods in adults with long dorsal lobes [31]. Furthermore, the recovery of the monophyly of Diplostraca through the exclusive use of ho-moplastic sites and the application of site-heterogeneous models (Figure 2a) confirmed that the non-monophyly of Diplostraca was an artifact. In FcLM analyses based on nucleotide sequences, the majority of quartets supported Notostraca as the closest relatives of Laevi-caudata (Figure 3). This is congruent with part of the current results (Figure 2a and Figure S1a). In FcLM analyses based on amino acid sequences, the majority of quartets supported Notostraca as the closest relatives of Onychocaudata (Figure 4). This is again congruent with part of our results (Figure 2b and Figure S1b). Using amino acid sequences or the removal of fast-evolving sites was considered an efficient approach to reduce systematic errors and to resolve deep relationships [60–62]. However, the quartet puzzling analysis plotted the probability of the preferred: (((Laevicaudata, Onychocaudata), Notostraca) topology, Anostraca) and the probabilities only ranged from 1.1% to 26.4% (Figures 3 and 4). Measurement of phylogenetic signal showed 0.8% of unresolved quartets and 13.1% of partly resolved quartets presented in the Paas7 matrix (excluding 46.45% sites), and quartet support for preferred topology was still low (23.2%). These results suggested that the phylogenetic signal for a deep relationship between Laevicaudata and Onychocaudata was always weak and differed among amino acid datasets (Figure 3). These findings could be explained by the decay of the phylogenetic signal or a limited signal in the mitogenomic sequences. The limitations of mitogenomes applied in deep phylogeny of Arthropod have already been pointed out [63] and emphasized [1,4–6,64]. When the non-phylogenetic sig-nal was higher than the phylogenetic signal due to mutational saturation, high AT-content, parasitic life-styles or explosive radiation events, considerable systematically erroneous relationships were recovered [6]. Our analyses confirmed these conclusions.

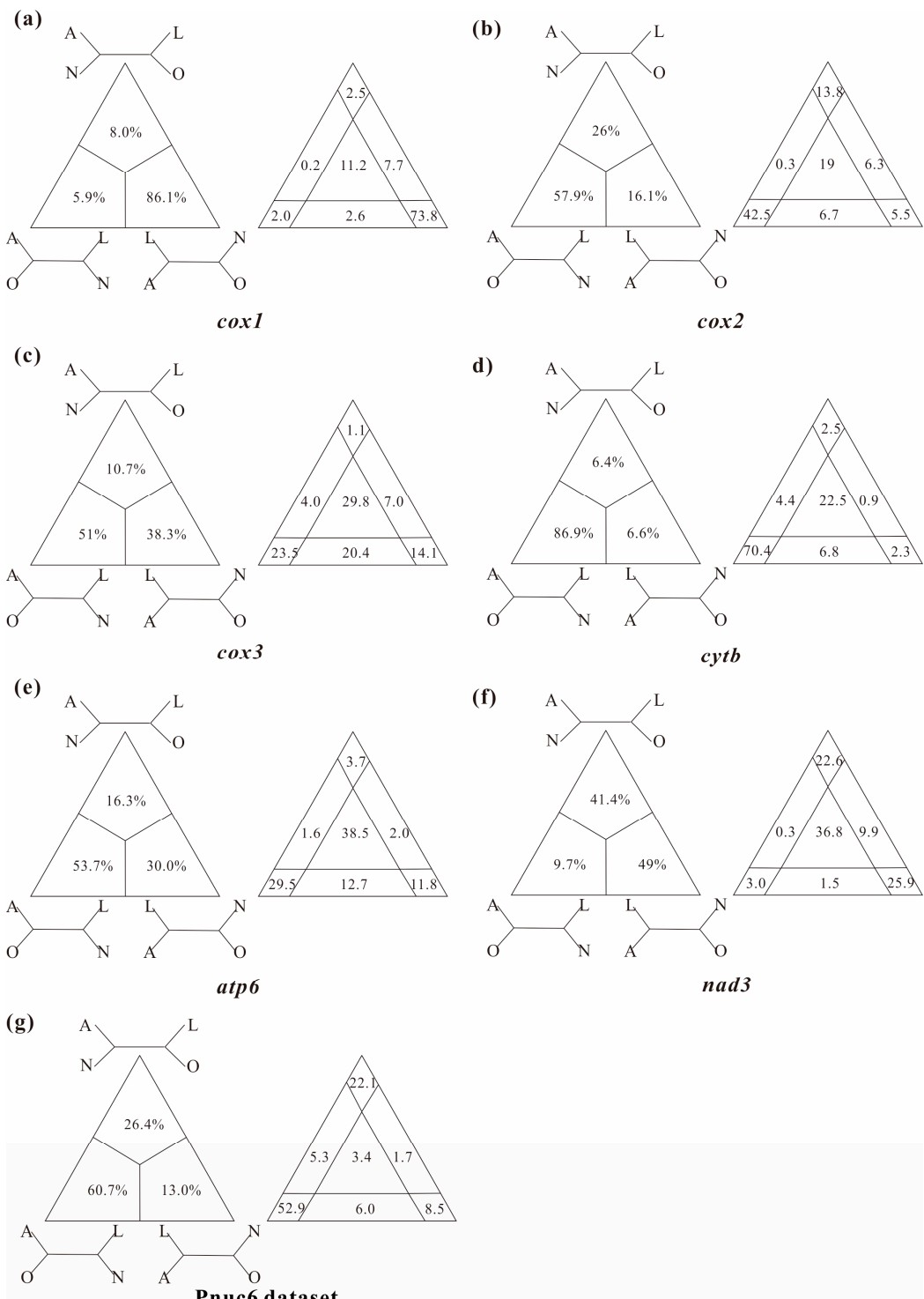

**Figure 3.** Conflict visualization using four-cluster likelihood mapping analyses as implemented in Treepuzzle based on the nucleotide sequence alignment of the first and the second codon positions of *cox1* (**a**), *cox2* (**b**), *cox3* (**c**), *cytb* (**d**), *atp6* (**e**), *nad3* (**f**), and Pnuc6 dataset (**g**). Quartet proportions (in %) are mapped on a 2D-simplex graph supporting different quartet topologies on the major phylogenetic relationships within Branchiopoda.

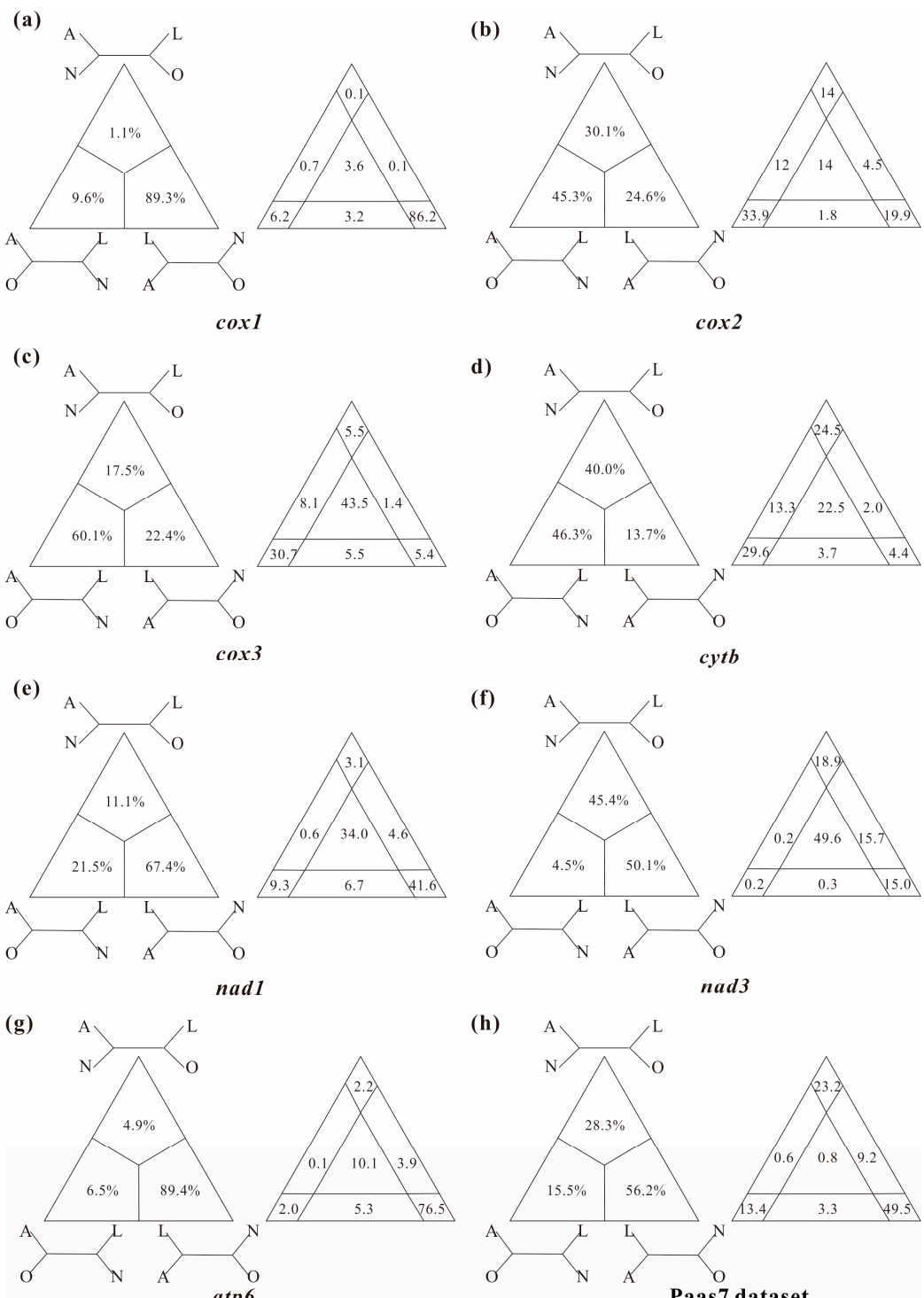

**Figure 4.** Conflict visualization using four-cluster likelihood mapping analyses as implemented in Treepuzzle based on the amino acid sequences of *cox1* (**a**), *cox2* (**b**), *cox3* (**c**), *cytb* (**d**), *nad1* (**e**), *nad3* (**f**), atp6 (**g**), and Paas7 dataset (**h**). Quartet proportions (in %) are mapped on a 2D-simplex graph supporting different quartet topologies on the major phylogenetic relationships within Branchiopoda.

### 4.2. Heterogeneity and Tree Topology

The comparisons of AT% between Laevicaudata and Spinicaudata (Table 2), the $I_D$ test (Table 3) and the PPA test (Table 4) demonstrated a high degree of compositional heterogeneity among branchiopod mitogenomes at both the nucleotide and amino acid

levels, which could bias the phylogenetic inference. The CAT+GTR model used in Bayesian inference analyses, implemented in PhyloBayes 4.1c [46], was chosen for its superiority in accommodating site-heterogeneous patterns of molecular evolution [11,12]. However, the Bayesian analysis under the site-heterogeneous model for the amino acid dataset (Paa7 matrix) recovered a topology almost identical to the phylogenetic analysis under the site-homologous model for the amino acid dataset of the PAA matrix (Figures 1b and 2b). This suggested that the homoplasy in amino acid datasets was not only due to compositional heterogeneity considered in site-heterogeneous model. When we applied amino acid recoding to our datasets (removing fast-evolving sites from Paa7 matrix; the Paas7 matrix) combined with the site-heterogeneous model, the monophyly of Diplostraca was recovered correctly (Figure 2c). Our study demonstrated that removing fast-evolving sites could be an effective method to overcome the among-site rate heterogeneity from nonstationarity [65,66]. Although phylogenetic resolution to the monophyly of Diplostraca was improved when we applied a variety of strategies to reduce the effects of saturation and heterogeneity, the deep relationships within Diplostraca were not fully resolved.

The sum of these analyses suggested that the phylogenetic resolution of Diplostraca using mitogenomes was trapped by conflicting phylogenetic signals existing across different genes, which in turn was aggravated by compositional heterogeneity and among-site rate heterogeneity. The phylogenetic signal and the potential influence of non-phylogenetic signal should be independently evaluated when mitogenomic datasets were applied in deep phylogeny.

## 5. Conclusions

In this study, we extensively dissected the potential sources of non-phylogenetic signal that resulted in high support but incorrect phylogenies when mitogenomes were applied in deep phylogeny. We identified significant compositional heterogeneity in both the nucleotide and amino acid datasets. Phylogenetic analyses under site-homogeneous models suggested that topological conflict at the base of Phyllopoda were retained across all datasets, even with the exclusion of the genes with the most strongly deviating compositions. Bayesian inference under the site-heterogeneous CAT-GTR+$\Gamma_4$ mixture model using the nucleotide dataset (Pnuc6) recovered the monophyly of Diplostraca. However, it is limited for the amino acid dataset, regardless of minimization of model violation. Although slow-evolving sites of the amino acid dataset (Paas7) under the site-heterogeneous model revealed the monophyly of Diplostraca with high support, the deep relationships among Laevicaudata, Spinicaudata and Cladocera were not fully resolved, which demonstrated systematic conflicts in phylogenetic signal. The results of FcLM analyses confirmed the systematic conflicts and revealed that the phylogenetic signal for deep relationship between Laevicaudata and Onychocaudata was significantly weaker than the nonphylogenetic signal across all datasets. Future analyses including the mitogenomes of the other laevicauatan species are needed to achieve a more complete understanding of the evolutionary history of Diplostraca by identifying more basal branches.

**Supplementary Materials:** The following supporting information can be downloaded at: https://www.mdpi.com/article/10.3390/cimb45020054/s1, Figure S1: Phylogenetic trees of Branchiopoda (colour-coded) obtained with RAxML and MrBayes inferred from the datasets of Pnuc6 (a) and Paa7 (b). Supports at nodes are ML bootstrap values and Bayesian posterior probabilities; Table S1: List of primer combinations used to amplify the mitochondrial genome of *L. grossipedia*; Table S2: Details of species and mitogenomes of Branchiopoda used in this study; Table S3: Partition schemes and best-fitting models for phylogenetic analyses; Table S4: Results of the test for substitution saturation. The references [67–87] are cited in the Supplementary Materials.

**Author Contributions:** Conceptualization, X.S. and J.C.; writing—original draft preparation, X.S.; writing—review and editing, J.C. All authors have read and agreed to the published version of the manuscript.

**Funding:** This work was supported by the National Natural Science Foundation of China (41730317, 42072022, 42288201), by the Chinese Academy of Geological Sciences (DD20221829), and by the State Key Laboratory of Palaeobiology and Stratigraphy at Nanjing Institute of Geology and Plaeontology, Chinese Academy of Sciences.

**Institutional Review Board Statement:** Not applicable.

**Informed Consent Statement:** Not applicable.

**Data Availability Statement:** All gene sequence data are available from GenBank (http://www.ncbi.nlm.nih.gov, accessed on 28 September 2022).

**Acknowledgments:** The authors are grateful to Shen YB for his encouragement and useful help.

**Conflicts of Interest:** The authors declare no conflict of interest. The funders had no role in the design of the study; in the collection, analyses, or interpretation of data; in the writing of the manuscript; or in the decision to publish the results.

## Abbreviations

*atp6*: ATPase subunit 6; *atp8*: ATPase subunit 8; *cox1*: Cytochrome c oxidase subunit I; *cox2*: Cytochrome c oxidase subunit II; *cox3*: Cytochrome c oxidase subunit III; *cytb*: mitochondria-encoded cytochrome B; *nad1*: NADH dehydrogenase subunit I; *nad2*: NADH dehydrogenase subunit II; *nad3*: NADH dehydrogenase subunit III; *nad4*: NADH dehydrogenase subunit IV; *nad4L*: NADH dehydrogenase subunit 4L; *nad5*: NADH dehydrogenase subunit V; *nad6*: NADH dehydrogenase subunit VI; tRNA: mitochondrial transfer RNA; *rrnS*: mitochondrial small ribosomal subunit RNA; *rrnL*: mitochondrial large ribosomal subunit RNA.

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
