# Peer review of "Conflicts in Mitochondrial Phylogenomics of Branchiopoda, with the First Complete Mitogenome of Laevicaudata (Crustacea: Branchiopoda)"

_cimb, doi:10.3390/cimb45020054_

Round 1

Reviewer 1 Report

The paper is well-done. Congadulations for the authors! The paper is bases on multy-step analysis which looks like modern. The illustrations are well-done. The authors are auto-critical in the Discussion of their results.

In few cases I feel not so perfect English, but I am not native speaker! Please show the MS to a native speaker.

Author Response

Response to Reviewer 1 Comments

Point 1: The paper is well-done. Congatulations for the authors! The paper is bases on multy-step analysis which looks like modern. The illustrations are well-done. The authors are auto-critical in the Discussion of their results.

In few cases I feel not so perfect English, but I am not native speaker! Please show the MS to a native speaker.

Response 1: Thank you for your praise of our work. We have submitted the revised manuscript to MDPI for professional language editing.

Reviewer 2 Report

Reviewer’s comment #: -The manuscript describes the Compositional heterogeneity and phylogenetic inference of Branchiopoda, with the first complete mitogenome of Laevicaudata (Crustacea: Branchiopoda). To overcome systematic bias the authors try to propose the most comprehensive analysis on the mitogenomic dataset which represents all major banchiopads. The title selected for the article is a bit broader than the data presented in it. The techniques used in the manuscript are up to date and the experiments are well performed. However, I found some parts and points, typos that needs correction and clarification to support and strengthen the final conclusion.

Abstract: Please rearrange the abstract for more clear information. There are many place where I cannot connect.

Introduction:

Few latest references (last 10 years study) are missing please incorporate them in support to the purpose of this study. The last paragraph of the introduction need to clarify the aim and novelty of this study.

Materials and methods:

Please incorporate the all the details of Sequence assembly and gene annotation which is almost missing.

Results and Discussion:

The figure representations are clear and well defined but the discussion supporting the results can be improved.

General Comments:

Since I found some degree of difficulty in reading and understanding certain parts of the manuscript, I feel this manuscript needs correction in the case of materials and methods and discussion. Overall there are many typos and grammatical corrections. I recommend go for English correction with the help of native speaker or through a company. I do think that the manuscript contains important issues, interesting approaches, which can lead to complete mitogenome of Laevicaudata. Therefore, I consider this manuscript suitable for publication after the suggested major revision in CIMB.

Reviewer 3 Report

The authors calculated the composition heterogeneity; reported the complete mitochondrial genome of Lynceus grossipedia Sigvardt et al., 2020, which was the first representative of the suborder Laevicaudata, and reconstructed the phylogeny tree. The findings from this study would further help in the evolutionary studies of other taxa. The manuscript is generally well written. However, there are some problems to be further improved as well. I would recommend accept with minor revisions.

Comment:

Line 16-17: “Phylogenetic relationships were investigated using several methods.”

How many methods? Please specify.

Line 44-65: The classification system used should be clarified at first.

Line 75: “Lynceus grossipedia” should be italic.

Line 89: Please add the longitude and latitude.

Line 95-100: The major procedures should be put here.

Line 128: The used software was PlyloBayesv4.1c, while the reference [39] is PhyboBayes 3. Please verify.

Line 130-131: Why chose these four clades: 1) Anostraca; 2) Notostraca; 3) Laevicaudata; and 4) Onychocaudata?

Line L145: In line 137, five datasets were used for phylogenetic analyses. Why four datasets here? Similarly, why three datasets were calculated in Line 159.

Round 2

Reviewer 2 Report

No comments